# Synthesis of Carbon-Supported PdIrNi Catalysts and Their Performance towards Ethanol Electrooxidation

**DOI:** 10.3390/mi12111327

**Published:** 2021-10-28

**Authors:** Ahmed Elsheikh, Hamouda M. Mousa, James McGregor

**Affiliations:** 1Chemical and Biological Engineering Department, University of Sheffield, Sheffield S1 3JD, UK; 2Mechanical Engineering Department, South Valley University, Qena 83523, Egypt; hmousa@eng.svu.edu.eg (H.M.M.); james.mcgregor@sheffield.ac.uk (J.M.)

**Keywords:** supported intermetallic nanoparticles, alloy formation, surface microstructure, ethanol oxidation

## Abstract

Direct ethanol fuel cells (DEFCs) have shown a high potential to supply energy and contribute to saving the climate due to their bioethanol sustainability and carbon neutrality. Nonetheless, there is a consistent need to develop new catalyst electrodes that are active for the ethanol oxidation reaction (EOR). In this work, two C-supported PdIrNi catalysts, that have been reported only once, are prepared via a facile NaBH_4_ co-reduction route. Their physiochemical characterization (X-ray diffraction (XRD), transmission electron microscopy (TEM), energy-dispersive X-ray spectroscopy (EDX), and X-ray photoelectron spectroscopy (XPS)) results show alloyed PdIrNi nanoparticles that are well dispersed (<3 nm) and exist in metallic state that is air-stable apart from Ni and, slightly, Pd. Their electrocatalytic activity towards EOR was evaluated by means of cyclic voltammetry (CV) and chronoamperometry (CA). Even though the physiochemical characterization of PdIrNi/C and Pd_4_Ir_2_Ni_1_/C is promising, their EOR performance has proven them less active than their Pd/C counterpart. Although the oxidation current peak of Pd/C is 1.8 A/mgPd, it is only 0.48 A/mgPd for Pd_4_Ir_2_Ni_1_/C and 0.52 A/mg_Pd_ for PdIrNi/C. These results were obtained three times and are reproducible, but since they do not add up with the sound PdIrNi microstructure, more advanced and in situ EOR studies are necessary to better understand the poor EOR performance.

## 1. Introduction

With the consequences of growing climate change and consistent worldwide energy demand, there is an urgent need to find alternative energy sources and technologies that are sustainable and friendly to the environment [1,2,3]. Fuel cells are one of the promising technologies which can contribute to energy demands while mitigating environmental challenges by reduction of greenhouse gas (GHG) emissions unlike conventional heat engines [4,5,6,7]. Direct application of ethanol into the fuel cell anode is a recent crucial direction in the research community. One of its advantages, as a liquid, ethanol is easy to store and transport. Moreover, it is energetic (8 kWh/L), and can be produced from biomass and agricultural sources [8,9]. Also, it is less likely than methanol to cross-over the membrane from the anode to the cathode. Furthermore, it is also less toxic than methanol. Consequently, the development of direct ethanol fuel cells (DEFCs) is a promising research goal [9,10,11,12]. Despite those advantages, there remain challenges regarding the commercialization of DEFCs. One is the scarcity of Pt reserves [13,14]. Applying an alkaline medium instead of acidic one has shown advantages for the overall fuel cell performance and cost because it improves the reaction kinetics, enables using less- or non-noble metal catalyst, and provides a less corrosive environment [9,15]. To prepare the fuel cell catalyst, an inert and high surface-area support material is usually required to disperse the small metal nanoparticles and maximize the reaction area. Carbon materials such as vulcan carbon, carbon nanotubes, nanofibers, and graphene are the most common support materials that have been studied [16,17,18,19,20,21,22,23]. Furthermore, achieving lower metal loading and consumption and attaining dispersed metal species are other benefits of using a carbon support [24].

Pt can be replaced by Pd which is very similar to Pt but more abundant in the Earth. Another advantage is the higher Pd tolerance towards catalyst poisons (e.g., CO_x_ species) while Pt is highly susceptible to poisoning and its functional life ends quickly [13,25,26,27]. Furthermore, adding a second metal as a cocatalyst is technoeconomically beneficial because it decreases the noble metal consumption and can modify the electronic and geometric properties of Pd nanoparticles [28,29,30]. Among the metals that could promote Pd electrocatalytic activity towards EOR are Ni, Sn, Ir, Au, and Cu [5,31,32,33,34]. However, even with the bimetallic benefits, still a further improvement in the performance and reduction in cost are necessary to enable wider exploitation of DEFCs. Some groups have investigated the synthesis of trimetallic systems instead of bimetallic ones in order to maximize the technoeconomic benefits of co-catalyst metals added to Pd or Pt [4,12,32,35,36,37,38,39,40]. Adding Ir [2,6,33] or Ni [10,30,41,42] as a cocatalyst into Pd/C is proven to be beneficial towards EOR. The hypothesis of this work is that adding both Ir and Ni to Pd/C would multiply the catalytic promotion towards EOR more than the single addition of Ir or Ni. As far as the authors are concerned, only one similar work [4] of PdIrNi/C trimetallic systems has been reported in which the PdIrNi/C has outperformed its Pd/C, PdIr/C, and PdNi/C counterparts towards EOR.

## 2. Materials and Methods

The synthesis of the anodic tri-metallic catalysts in the present study followed the previously reported works [33,43,44] with minor modification. All chemicals were purchased from Sigma-Aldrich (Gillingham, UK) except vulcan carbon which was purchased from Capot Corp (Boston, MA, USA). Table 1 shows the stoichiometric added quantities of metal and carbon precursors to each respective sample. To prepare the catalysts, the respective metal and carbon precursors were mixed in a mixture of 2-propanol and water (50/50, *v*/*v*). Then, KBr was added as a capping agent with an atomic KBr/metal ratio of 1.5:1. The mixture was stirred for 20 min before the NaBH_4_ solution (0.5 M, 15 mL) was added under stirring. It was, subsequently, kept under stirring for 30 min. Finally, the catalyst was copiously washed in deionized water and vacuum filtered and dried in a vacuum oven at 120 °C for 2 h.

The crystal structure was examined by X-ray diffraction (XRD) utilising a Bruker D2 Phaser (Billerica, MA, USA) operating at 30 mA and a scan rate of 12°/min. Transmission electron microscopy (TEM) was performed to study the surface morphology, particle shape, and size distribution using a Philips/FEI CM 100 microscope (Hillsboro, OR, USA) operating at 100 kV with a LaB6 filament. The TEM samples were prepared by spraying 20 µL of the catalyst slurry (powder + ethanol) over the C-coated Cu grids which were then left to dry overnight. The elemental composition was examined by an X-ray spectroscopy (EDX) detector attached to a JEOL 6010A scanning electron microscope (Akishima, Tokyo, Japan). Two accelerating voltages of 10 kV and 20 kV were applied to investigate the composition at varying depths from the surface [45]. The valence state and top-surface composition were examined by X-ray photoelectron spectroscopy (XPS) using a Thermo Fisher Scientific K-alpha + spectrometer (Waltham, MA, USA). A monochromatic Al X-ray source (72 W) was used to analyse samples over 400 µm^2^ of area. Data were recorded at 120-eV and 40-eV pass energies for the survey and high-resolution scan, respectively. Data analysis was performed in CasaXPS from Casa Software Ltd. (Teignmouth, UK) and using a Shirley type background and Scofield cross sections, with an energy dependence of −0.6. To evaluate the catalyst performance towards ethanol electrooxidation, cyclic voltammetry (CV) and chronoamperometry (CA) tests were performed in a home-made three-electrode half-cell. A Gamry reference 600 mini-station from Gamry Instruments Inc. (Warminster, PA, USA) was used to evaluate the performance. The working electrode was a glassy carbon electrode (ϕ 3 mm) onto which the catalyst ink slurry (25 µL) was drop-casted and dried. That slurry was prepared by dispersing 5 mg of each catalyst powder in a mixture of ethanol (2 mL) and Nafion^®^ 117 5 wt.% (25 µL) followed by 1 h of stirring. The reference and counter electrode were Ag/AgCl (sat’ KCL) and Pt wire, respectively. The reference electrode potential was converted to normal hydrogen electrode (NHE) in the voltammograms. To facilitate the reactant mass diffusion across the electrode-electrolyte interface, the electrochemical tests were performed in magnetically stirred solutions at a speed of 50 rpm.

## 3. Results

Figure 1 shows the XRD patterns of PdIrNi/C, Pd_4_Ir_2_Ni_1_/C, Pd/C, Ni/C, and Ir/C. The broad peak at ~25° is attributed to the semi-crystalline nature of graphitic vulcan carbon. The basal C (002) peak is broader than the other metallic peaks in Pd/C, Ir/C, Pd_4_Ir_2_Ni_1_/C and PdIrNi/C. However, it is more intense in the case of Ni/C, probably because small-size oxidized Ni species are the main constituent, but the peak at approximately 43° seems to represent the *fcc* metallic Ni peak of (111) according to [30,46]. The other peaks of Ni/C at ~35°, and ~60° are attributed to nickel hydroxide species according to the same references. Considering the Pd/C pattern, this shows another three peaks located at approximately 40°, 46°, 67.8°, and 82.3° ascribed to the Pd polycrystalline phases of Pd (111), Pd (200), Pd (220), and Pd (311), respectively. Similar peaks to those exist—but positively shifted 1° in the case of Ir/C—due to the smaller crystal lattice size and atomic radius of Ir compared to Pd.

As for PdIrNi/C and Pd_4_Ir_2_Ni_1_/C catalysts, it can be noted that the presence of Ir and Ni has shifted the XRD peaks to higher angle values towards pure Ir pattern. This shift is demonstrated in the inset graph in Figure 1 which shows the PdIrNi is reflected at angles close to those of pure Ir which was also noted in [4,33]. According to [33], the lower *d*-space Ir (d_111_ = 2.217 Å) could be incorporated into the larger *d*-space Pd (d_111_ = 2.217 Å) and form an alloy regardless of their atomic proportions. The crystal lattice of Pd, therefore, might be subjected to compressive stresses and strains that have resulted in lattice contraction in the trimetallic samples. Also, peak broadening in the case of PdIrNi/C is higher than that of Pd/C and Ir/C which could be explained by the decreasing particle size. Notably, the Ni(OH)_2_ peaks at 35° and 60° that have been reported in Ni-containing trimetallic and bimetallic catalysts [30,46] are suppressed in the PdIrNi/C diffractogram. This implies a good overall mixing of the three metals and that an alloy of Pd, Ir, and Ni was formed. The crystallite size (*τ*, nm) is estimated using the Scherrer’s Equation (1) and the catalyst respective sizes are listed in Table 2 which shows a significant particle size reduction from 4 nm for the monometallic Pd/C to 1.4 and 1.8 nm for PdIrNi/C and Pd_4_Ir_2_Ni_1_/C, respectively.
(1)$$\tau =\frac{K\lambda }{\beta \cos\left(\theta \right)}$$
where, *K* is constant (0.94), *λ* is wavelength of Cu and equals 0.154 nm, *β* is the is the full width at half-maximum height in radians, *θ* is half of the diffraction angle. Table 2 also lists the estimated lattice constant for Pd/C, PdIrNi/C, and Pd_4_Ir_2_Ni_1_/C. The pure Pd lattice constant is approximately 3.89 Å [47], but as shown in Table 2, the C-supported Pd lattice constant is 0.11-Å larger. This expansion in the Pd lattice is possibly due to the interaction between Pd and C support during synthesis. This interaction exercises a tensile strain on the Pd crystal lattice. Adding Ni and Ir in PdIrNi/C leads to 0.3-Å reduction in the lattice constant which could be attributed to the replacement of some Pd atoms by Ir and Ni ones in the formed alloy. While maintaining the metal loading rate 12 wt.%, decreasing the Ir and Ni content added to Pd in Pd_4_Ir_2_Ni_1_/C results in less reduction (0.2 Å) of the Pd constant lattice which, also, could be the result of Ir and Ni atoms replacing Pd ones. According to [47], the lattice constants of Ir and Ni are 3.84 and 3.52 Å, respectively.

To investigate the composition of the trimetallic system and metal load on carbon, energy-dispersive X-ray spectroscopy (EDX) was performed at two different accelerating voltages aiming to vary to the depth at which the composition is analysed [45]. Table 3 shows the weight concentration of each metal at 10 kV and 20 kV. The metal loading values are close but slightly higher that the nominal loads of 12 wt.%. This might be attributed to two reasons: first the electron beam used in EDX analysis does not travel through the sample but is reflected after penetrating a certain depth from the top surface and, therefore, does not analyse the bulk composition [45]. Secondly, the metal particle surface energy could have exercised a segregation potential which is more obvious in the higher Ni-containing PdIrNi since Ni has a significant tendency to segregate into the surface [42]. Furthermore, both samples metal load at 10 kV is higher than that at 20 kV which incorporates the metal nanoparticle tendency to segregate into the surface.

Figure 2 shows the EDX elemental spectrum of PdIrNi at 20 kV. The predominantly high C–K peak is not shown in order not to suppress the other metallic ones. The peaks of Ir–M, Pd–L, and Ni–K are all visible in the spectrum. The Na, S, and Si peaks could be ascribed to the Vulcan carbon impurities. The O–K one is probably due to either the carbon functional groups or Ni oxide species. It is noteworthy that Ni concentration at 10 kV is almost twice its concentration at 20 kV for both trimetallic samples; a trend that is not noted for both Pd and Ir. This finding is crucial because it proves the high surface segregation potential of Ni [42,46]. Another important observation from Table 3 is that both Pd and Ir concentrations in both samples measured at 10 kV are higher than those at 20 kV, but the Ir increase is higher than that of Pd which shows the Ir’s higher tendency of surface segregation. Figure 3 shows the EDX elemental maps of PdIrNi/C and Pd_4_Ir_2_Ni_1_/C recorded at 20 kV. The Ir and Pd maps of PdIrNi/C (Figure 3A,B) show more densely populated samples as compared to Ni (Figure 3C). The less dense population of Ni map is due to the lower atomic Ni concentration detected compared to Pd and Ir (Table 3) even though the nominal added quantities of the three metals are equivalent. This is probably because Ni tends to segregate to a higher surface level above the EDX electron beam interaction volume which is supported by the significant XPS surface concentration (Table 4) of Ni (1.25 at.%) that is higher than Pd (1.21 at.%) but lower than that of Ir (1.48 at.%). A similar trend could be observed for Pd_4_Ir_2_Ni_1_/C sample (Figure 3D,E,G).

To study the nanocatalyst surface morphology, transmission electron microscopy (TEM) was performed. Figure 4 shows the micrographs and particle size distribution of Pd/C (A), PdIrNi/C (B) and Pd_4_Ir_2_Ni_1_/C (D). For Pd/C, dispersed metal dark particles (2–10 nm) appear along larger grey carbon aggregates (30–60 nm). Nonetheless, particle agglomeration could be seen for this sample. Intense aggregated particles are visible in the middle area but this is, probably, due to the TEM sample preparation and not something inherent in the Pd/C powder. The particle size distribution of that sample shows the average particle size (of approximately 100 particles) equals 5 nm which is close to but larger than the XRD size (Table 2). Figure 4B shows the micrograph and particle size distribution of PdIrNi/C which shows enhanced particle dispersion compared to Pd/C (Figure 4A). The average particle size of PdIrNi is 1.9 nm which is 3.1 nm smaller than Pd. Likewise, the particle size of Pd_4_Ir_2_Ni_1_ (Figure 4C) equals 2.3 nm which is 2.7 nm less than Pd but 0.4 nm larger than PdIrNi. The TEM analysis showed significant particle size reduction in the trimetallic samples compared to Pd/C. This is a usual finding when preparing bimetallic and trimetallic supported nanoparticles due to difference in metal chemistries [41,48,49]. Another reason for the trimetallic particle size decrease of PdIrNi is that since the geometrical parameters of Ir and Ni are smaller than those of Pd, they are likely to be incorporated into the latter’s lattice exercising compressive stress and strain causing a net lattice contraction. As shown in Figure 1, adding Ir to Pd shifts its diffraction angles to higher values and exercise compressive strain on it. The increase of particle size of Pd_4_Ir_2_Ni_1_ compared to PdIrNi is, probably, due to the decreased proportion of Ni and Ir and increased one of Pd in the former since the overall metal load is unchanged.

The electrocatalyst dispersion (% *N_S_*/*N_T_*) could be estimated by applying the Van del Klink Equations (2)–(4) [42]:(2)$$N_{T}=\frac{2\pi }{3}\times {\left(\frac{d}{a}\right)}^{3}$$
(3)$$N_{T}=\frac{10}{3}l^{3}-5l^{2}+\left(\frac{11}{3}\right)l-1$$
(4)$$N_{S}=10l^{2}-20l+12$$
where, *N_T_* is the total number of atoms, *d* is the average TEM particle size (nm), *a* is the lattice constant, *l* is the number of layers and *N_s_* is the number of surface atoms. As listed in Table 2, the lowest dispersion is that of Pd/C followed by Pd_4_Ir_2_Ni_1_/C and PdIrNi/C is the highest in particle dispersion as listed in (Table 2).

To study the valence state and further examine the surface composition, XPS analysis was performed. Figure 5 shows the full XPS surveys of Pd/C, Pd_4_Ir_2_Ni_1_/C, and PdIrNi/C. The global and most intense peak at 284 eV represents the carbon 1s peak. Additionally, the peaks of Ir 4f, Pd 3d, and Ni 2p are located at approximately 62, 335, and 858 eV, respectively. Figure 6C shows the enlarged C1s peak which shows the predominant component of vulcan carbon is hybridised sp^2^. In the three sample spectra, there is an overlap between the peaks of Pd 3d and O 1s which is shown in Figure 6C. Table 4 shows the atomic concentration of each metal and its oxide form in addition to the binding energy value that is distinctive to each metal. Also, the details of elemental peaks of Pd, Ir, and Ni in the three samples are shown in Figure 6. The Pd 3d peak of Pd/C is shown in Figure 6A and it is deconvoluted into high- and low-energy bands. The lower-energy Pd 3d_5/2_ is located at 335.43 which is positively shifted 0.4 eV compared to pure Pd due to the interaction with the carbon support. This is even further shifted to a higher value (335.54) upon addition of Ir and Ni. It is noteworthy that some Pd exists in an oxidised form in both Pd/C and PdIrNi/C even though the oxide quantity is three times lower in case of PdIrNi/C. These signals that adding Ni and Ir has enhanced the Pd air stability. The majority (75%) of nickel, however, exists in oxidised form and the remaining 25% are metallic. The Ir exists only in metallic form without any oxide unlike the finding reported by [33] who prepared PdIr/C. That could explain why the Ni presence has enhanced the air stability of both Pd and Ir.

Figure 7A shows the cyclic voltammetry (CV) graphs of Pd/C, Pd_4_Ir_2_Ni_1_/C and PdIrNi/C in 1 M KOH recorded at 50 mV/s. The voltammogram of Pd/C is typical of other Pd ones reported [6,10,33,50]. The H adsorption/absorption is more pronounced on the trimetallic samples while it is largely suppressed on the Pd/C due to the capacity of Pd to absorb hydrogen into its core structure instead of undergoing surface reaction. Considering PdIrNi and Pd_4_Ir_2_Ni_1_, Ir behaves similarly to Pt since it adsorbs H on its surface [6,33]. Therefore, the higher current noted around −600 mV is ascribed to that process. Upon increasing the applied potential further, OH adsorption commences around −300 mV. It is noteworthy that Pd/C shows enhanced adsorption compared to the trimetallic samples, probably due to the lower capability of Ir and Ni to adsorb OH [33]. The OH adsorption is, technically, the onset of surface oxidation but it continues until the actual surface oxidation starts around 50 mV, which continues until the end of the forward scan. However, a sharp increase in the current on PdIrNi/C is noted, at the that end, which is due to Ni(OH)_2_ oxidation to NiOOH which is then reduced back to Ni(OH)_2_ in the reverse scan around 350 mV [42]. The Ni(OH)_2_-associated current increase is much less on Pd_4_Ir_2_Ni_1_/C than that of PdIrNi/C which is probably due to the significantly lower Ni quantity in the former. Similarly, the NiOOH-associate current decrease was noted in the reverse scan on Pd_4_Ir_2_Ni_1_/C for the same reason.

After adding ethanol to KOH (Figure 7B), the H_ads/abs_ peak was largely suppressed by the ethanol adsorption. At slightly higher potential (−500, −400 mV), OH adsorption commences, facilitating the oxidation of adsorbed ethoxy species, releasing an increasing current with increasing the applied potential (increasing the adsorbed OH). The single C-supported Ir shows no activity towards EOR according to [6]. However, in the same article, adding Ir to Pd/C enhanced the EOR kinetics which the authors attributed to water activation. This trend continues until Pd surface oxidation occurs at −50 mV for PdIrNi/C and +50 mV for Pd/C. A similar phenomenon is noted on Pd_4_Ir_2_Ni_1_/C. Above such potentials, no more OH is adsorbed on Pd as the Pd sites are oxidised which decreases the overall current. It is noteworthy that the forward current density obtained with PdIrNi/C and Pd_4_Ir_2_Ni_1_/C is significantly less than that of Pd/C. This is the opposite of the expected benefits of Ni [41,51,52] and Ir [2,6,53] upon their individual addition to Pd/C towards EOR. As for the reverse scan of Pd/C, it can be noted that there are two small shoulder peaks between 300 and 100 mV and one highly intense peak at −150 mV. The latter is usually observed and attributed to removing the incompletely oxidised species from Pd active sites. Shoulder peaks are reported in the literature and are ascribed to the removal of reaction intermediates on the Pd surface due the dynamic stirring of the solution. The reverse peak is very broad on PdIrNi/C and Pd_4_Ir_2_Ni_1_/C.

Considering PdIrNi/C, it can be noted that the high current at the forward scan end (1250 mA/mg_Pd_) is due to NiOOH ethanol activating activation as was found by Barbosa et al. [54]. Comparing this value to the current value in KOH only (225 mA/mg) illustrates that ethanol is oxidised by the oxidised Ni species. However, it is noteworthy that the upward going current density suggests that EOR on NiOOH has recently started at approximately 500 mV as shown in Figure 7B. A current density peak could be expected at higher positive potential as was found by [54] who also found two reduction peaks in the reverse scan, one of which was due to NiOOH backward reduction to Ni(OH)_2_. A similar conclusion was obtained regarding EOR on Ni(OH)_2_ microspheres by Lidasan et al. [55]. A further similar conclusion about NiO was obtained by Amin et al. [56].The overall oxidation current of PdIrNi/C is lower than that of Pd/C. This could be due to the inactivity of Ir and Ni to ethanol oxidation while they occupy most of the PdIrNi/C near-surface layers. Additionally, the positive Pd 3d binding shift (Table 4) is suggestive of stronger bonds between Pd and the poisons such as CO_x_.

To investigate the catalyst’s EOR stability, 2-step chronoamperometry scan was undertaken on each catalyst (Figure 8). The first constant potential was −0.3 V and was applied for 30 min. The second was +0.1 V and was applied for another 30 min starting immediately after the first step ended. The scans were performed in 1M KOH + C_2_H_5_OH. The −0.3 V potential, according to Figure 7, is located in the middle of OH adsorption potential window. Therefore, in Figure 8, a high tolerance for poisoning species is attained by the constant generation of adsorbed OH species that are capable of removing the adsorbed ethoxy. The higher current obtained for Pd/C that is more than the two trimetallic catalysts is probably because of the Pd surface abundance on it. This is further proved by the higher current density of Pd_4_Ir_2_Ni_1_/C than PdIrNi/C because the former’s surface contains Pd more than Ir and Ni combined while the latter’s surface contains equivalent molar concentrations of each metal (Table 3 and Table 4). Also, there is a shift to higher binding energies (approximately +0.1 eV) of Pd 3d according to the XPS (Table 4). Such a shift to a higher binding energy could be a negative indication of lower catalytic performance. A higher bonding strength existing between Pd sites and poisoning species on the trimetallic samples could present higher activation barriers of EOR compared to Pd/C. At −0.1 V, however, the initially remarkable current density, on Pd/C, decays rapidly with the time. The quick current deterioration is probably because of lacking OH species (Figure 6 and Figure 7) on the catalyst surface to remove the poisoning species which leads to permanent losses of Pd active sites. Nonetheless, the trimetallic samples achieve identical CA current densities—which are much less than Pd/C—at +0.1 V. That is once again because of less Pd surface presence. It is, also, noteworthy, that both catalysts achieve identical current density in the voltammogram forward scan (Figure 7) which is probably why they achieve identical CA current densities. However, both of them achieve stable current compared to Pd/C which is due the Ir and Ni potential to generate oxygen species that could remove the ethoxy from Pd active sites.

## 4. Discussion

The electrochemical test results show that both of the currently prepared PdIrNi/C samples have underperformed their monometallic counterpart towards EOR even though their physiochemical characteristics predict otherwise. Nonetheless, the results have been reproduced three times for each catalyst and the same performance was obtained that demonstrates such outcomes are reproducible. There are no solid explanations or understanding of what is actually happening with these samples during EOR. The baseline interpretation is that both trimetallic surfaces contain less Pd active sites than Pd/C which is not entirely inaccurate according to the EDX and XPS results although the exact in situ reduction of Pd active sites is unknown. Moreover, the exact consequent physical and electrochemical impact on the EOR mechanism is also unknown. More advanced analytical and in situ studies of EOR could reveal information that is not known, presently. Nonetheless, there is a previous PdIrNi/C report on EOR by Shen et al. [4] in which it outperformed its monometallic Pd and bimetallic PdIr and PdNi counterparts. The synthesis method, physiochemical characteristics, and testing conditions in the current work and [4] are similar to a large extent. The only notable differences are the stabilising agent and Ir and Ni content used. They used a potassium citrate as a stabilising agent, while KBr was used in this work. Since KBr was also used as a stabiliser in the synthesis of C-supported PdAgNi [57] that outperformed the monometallic Pd/C for EOR, it is unlikely to have a detrimental effect during EOR on PdIrNi. Another reason to think KBr does not have a negative effect is that it was, also, applied for C-supported PdAuNi that outperformed its monometallic Pd counterpart for direct borohydride elctrooxidation [58]. The other notable difference is the atomic ratio of Pd, Ir, and Ni which is 7:1:12 in [4] while the ratios in this work are 1:1:1 and 4:2:1. The Ir content in their sample is much lower than that at the current two samples and the opposite is true about Ni. How such different Ni and Ir contents might have impacted PdIrNi EOR performance is beyond the scope of this work.

## 5. Conclusions

Trimetallic C-supported PdIrNi/C nanoparticles have been successfully prepared by NaBH_4_ co-reduction. The XRD, TEM, EDX, and XPS results prove the formation of alloyed and well-dispersed (<3 nm) PdIrNi/C nanoparticles in both trimetallic samples existing in proportionate metallic state (apart from Ni). However, both trimetallic samples have underperformed their monometallic counterpart for ethanol electrooxidation. Given the good physicochemical characteristics, the exact mechanism through which PdIrNi catalysts have underperformed their Pd counterpart towards EOR is still unknown. Further analytical and in situ studies might shed more light on that. Another finding is that Ni is active for EOR but at significant overpotential compared to Pd, unfortunately.

## Figures and Tables

**Figure 1 micromachines-12-01327-f001:**
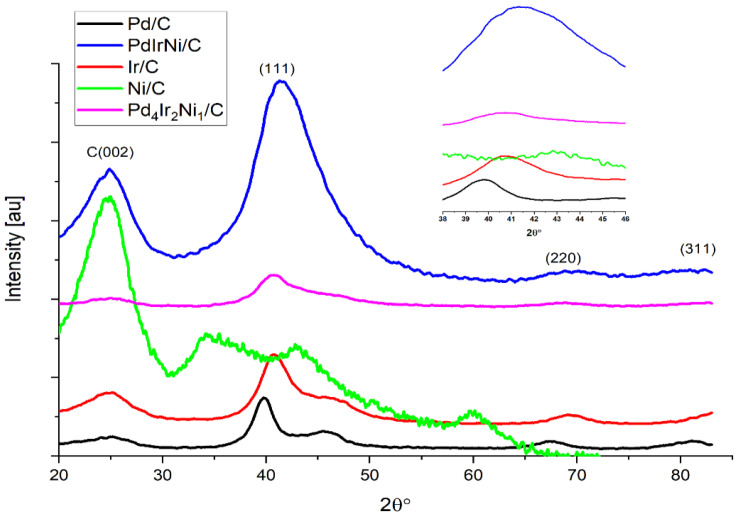
X-ray diffraction (XRD) patterns of Pd/C, Ir/C, Ni/C, Pd_4_Ir_2_Ni_1_/C and PdIrNi/C, inset figure: an enlarged (111) peak.

**Figure 2 micromachines-12-01327-f002:**
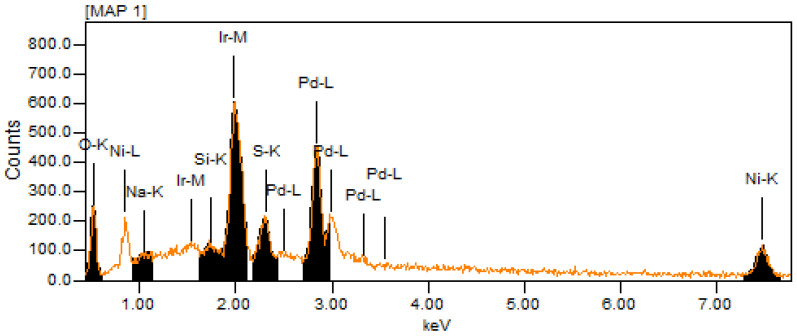
EDX elemental spectrum at 20 kV of PdIrNi.

**Figure 3 micromachines-12-01327-f003:**
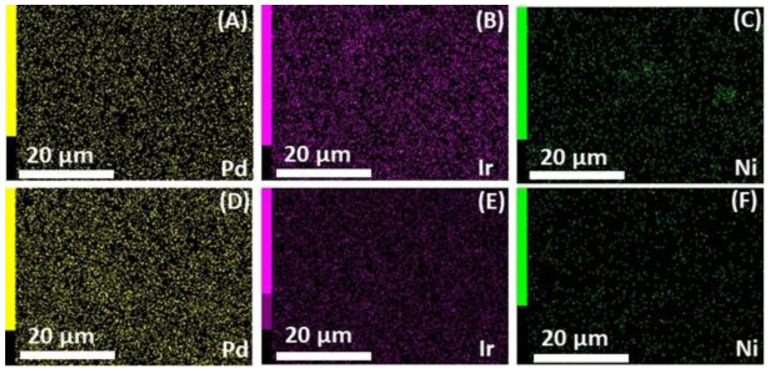
EDX Pd, Ir, and Ni maps of PdIrNi/C (**A**–**C**) and Pd_4_Ir_2_Ni_1_/C (**D**–**F**).

**Figure 4 micromachines-12-01327-f004:**
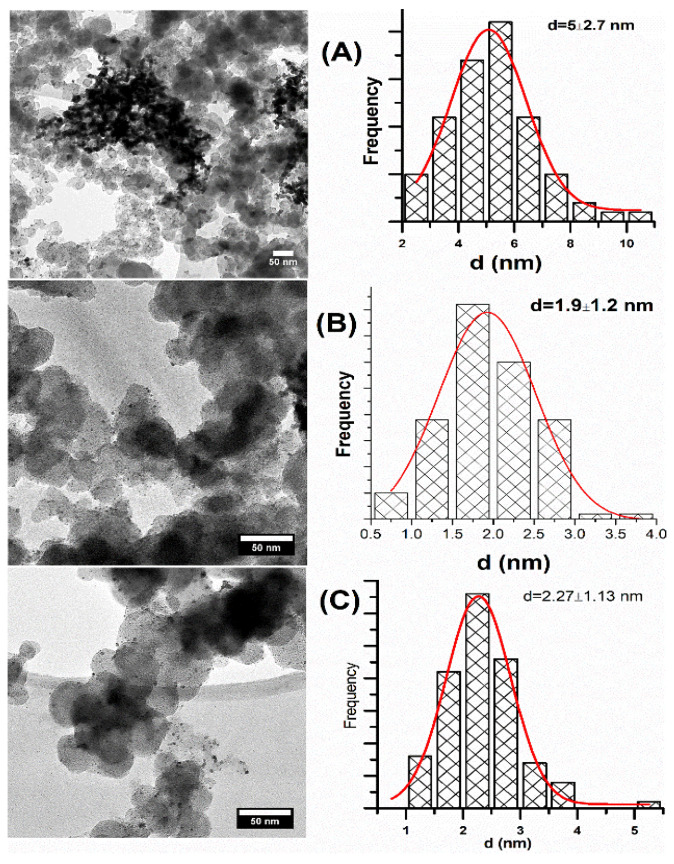
TEM micrographs and particle size distribution of Pd/C (**A**), PdIrNi/C (**B**), Pd_4_Ir_2_Ni_1_/C (**C**).

**Figure 5 micromachines-12-01327-f005:**
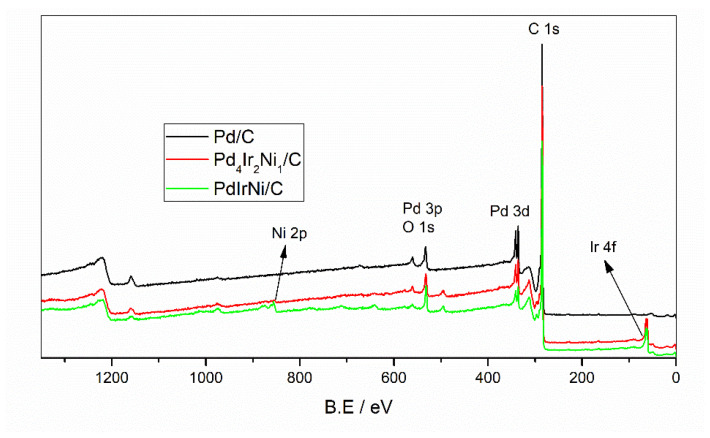
XPS full surveys of Pd/C, Pd_4_Ir_2_Ni_1_/C, and PdIrNi/C.

**Figure 6 micromachines-12-01327-f006:**
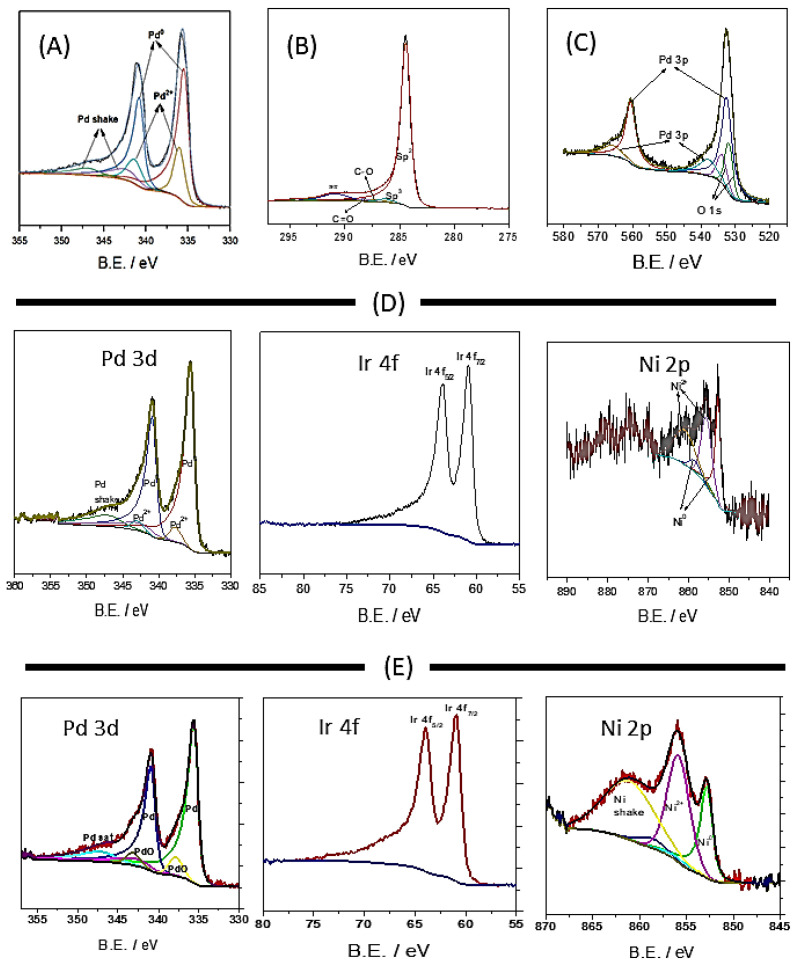
XPS peaks of Pd 3d (**A**), C 1s (**B**), Pd 3p + O 1s (**C**) in Pd/C, peaks of Pd 3d, Ni 2p, Ir 4f of Pd_4_Ir_2_Ni_1_/C (**D**) and PdIrNi/C (**E**).

**Figure 7 micromachines-12-01327-f007:**
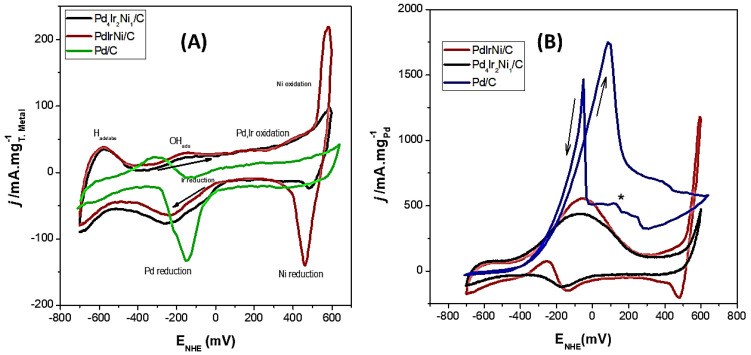
Cyclic voltammetry (CV) of Pd/C, PdIrNi/C, and Pd_4_Ir_2_Ni_1_/C in 1M KOH at 50 mV/s vs. NHE in 1M KOH (**A**) and 1M KOH + EtOH (**B**) (*) unknown peaks.

**Figure 8 micromachines-12-01327-f008:**
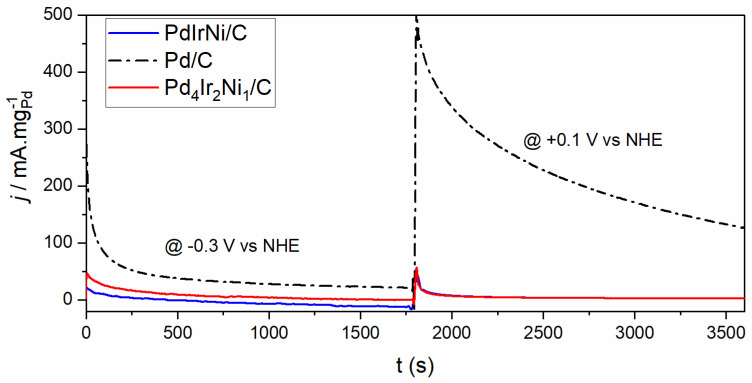
Step chronoamperometry (CA) scans of Pd/C Pd_4_Ir_2_Ni_1_/C and PdIrNi/C in 1M KOH + EtOH at −0.3 V and +0.1 V vs. NHE.

**Table 1 micromachines-12-01327-t001:** Nominal added metal and carbon precursors to prepare Pd/C, PdIrNi/C, and Pd_4_Ir_2_Ni_1_/C.

Catalyst	C (mg)	PdCl_2_ (mg)	NiCl_2_ (mg)	IrCl_3_ (mg)
Pd/C	132	29.4	-	-
PdIrNi/C	132	8.83	6.46	14.94
Pd_4_Ir_2_Ni_1_/C	132	14.19	2.58	11.94

**Table 2 micromachines-12-01327-t002:** Crystallographic and tomographic information of Pd/C, PdIrNi/C, and Pd_4_Ir_2_Ni_1_/C.

Catalyst	XRD * Size (nm)	Lattice Constant (Å)	TEM ** Size (nm)	% Dispersion
Pd/C	4	4	5	25
PdIrNi/C	1.4	3.7	1.9	52
Pd_4_Ir_2_Ni_1_/C	1.8	3.8	2.3	44

* XRD: X-ray Diffraction. ** TEM: Transmission Electron Microscopy.

**Table 3 micromachines-12-01327-t003:** Elemental energy-dispersive X-ray spectroscopy (EDX) compositional analysis and metal loading (wt.%) of the PdIrNi/C and Pd_4_Ir_2_Ni_1_/C surfaces.

Catalyst	Acc. Voltage	Pd		Ni		Ir		Metal Load (wt.%)
		wt.%	at.%	wt.%	at.%	wt.%	at.%	
PdIrNi/C	10 kV	5.75	0.41	4.86	1.14	7.40	0.41	18
	20 kV	5.36	0.62	2.51	0.53	6.62	0.39	14.5
Pd_4_Ir_2_Ni_1_/C	10 kV	7.15	0.97	1.15	0.28	6.95	0.52	15.3
	20 kV	6.64	0.90	0.56	0.14	6.09	0.46	13.3

**Table 4 micromachines-12-01327-t004:** X-ray photoelectron spectroscopy (XPS) data of valence state and metal/oxide atomic concentration measurements in Pd/C, Pd_4_Ir_2_Ni_1_/C and PdIrNi/C.

Catalyst	Pd at. %		Pd 3d_5/2_ (eV)	C at. %	Ir at. %	Ir 4f_7/2_ (eV)	Ni at. %		Ni 2p_3/2_ (eV) *
	Pd^0^	Pd^2+^					Ni^0^	Ni^2+^	
Pd/C	1.63	0.45	335.43	96.13	x	X	X	x	x
PdIrNi/C	1.18	0.13	335.54	91.09	1.48	60.88	0.32	0.93	852.74
Pd_4_Ir_2_Ni_1_/C	1.39	0.10	335.51	93.62	1.16	60.98	0.12	0.18	852.6

* The location of Ni^2+^.

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
