# Peer review of "Synthesis of Carbon-Supported PdIrNi Catalysts and Their Performance towards Ethanol Electrooxidation"

_micromachines, 2021, doi:10.3390/mi12111327_

Round 1

Reviewer 1 Report

See the attachment.

Reviewer 2 Report

MICROMACHINES

Manuscript Number: 1396971

Title: Synthesis of carbon-supported PdIrNi catalysts and their performance towards ethanol electrooxidation

In the current work, PdIrNi/C catalysts with two different molar ratios were synthesized by a facile NaBH4 co-reduction route and tested for their EOR activity. Although the physicochemical characterization (XRD, TEM, EDX, and XPS) of the trimetallic catalysts were evaluated as promising by the authors, their activities towards EOR are found significantly lower than this of monometallic Pd/C. Specifically, the ethanol oxidation current peaks of Pd/C, PdIrNi/C and Pd4Ir2Ni1/C are 1.8 A/mgPd, 0.52 A/mgPd and 0.48 A/mgPd, respectively. The authors concluded that the high molar concentrations (>50 at.%) of Ni and Ir in PdIrNi systems are disadvantageous for ethanol electrooxidation. Nevertheless, they highlighted that Ni(OH)2 has a high potential of activating EOR but unfortunately at significant overpotentials.

In my opinion, the authors performed interesting physicochemical but not enough electrochemical (activity and stability) characterizations. The second part of the used title it is not respected (….their performance towards ethanol electrooxidation). I hardly can find apparent innovation and obvious progress in this manuscript. In addition, the next drawbacks must also receive special attention. Thus, I cannot recommend this work for publication in Micromachines under the current status.

  • To further improve the quality of the paper, the novelty of this work should be clearly pointed out, taking into account works published at least the last decade. Please better explain which is the novelty herein. Combinations of Pd with Ir and Ni supported on carbon have been thoroughly investigated for their EOR activities in alkaline media in many works over the past decade, and specifically the trimetallic PdIrNi/C system. The authors refer to these works in the introduction section; however, they do not highlight the novel aspect of their current research relative to the already published similar works. The preparation of the PdIrNi catalyst (high molar ratios of Ir and Ni to Pd have already been investigated in works dealing with bimetallic Pd alloys)? The preparation method used (NaBH4 co-reduction procedure is an acknowledged synthesis method)? The high activity/stability? (Less than that of Pd/C).
  • The authors should clearly point out their contribution in EOR science, (and after comparing their results with those already existing in the literature) and put in evidence the advantages and the disadvantages of their catalyst. Under the current status, the contribution of the work is doubtful. In all the previous works (at least eight), the addition of Ir and Ni to Pd has been proven to beneficially effect the EOR activity, even for high atomic ratios of the added metals to bimetallic and trimetallic Pd alloys. In these works, in addition to the CV experiments, the conclusions were reached after a detailed description of the possible ethanol oxidation reaction pathways for the Pd alloys, or after applying several measurements and methods, such as DE-PEMFC single tests, DFT calculations, etc. In my opinion the argument of the current works, which is in opposition with the all the similar works already published, cannot stand validly only based on the evaluation of EOR by the basic electrochemical methods (CV and CA). The authors must provide much more evidence to defend their conclusions against so many other published works.
  • Please prepare a Table listing the best performing catalysts including more data (EOR rates, current densities, etc.) of works appeared the last two decades in International Literature. Compare them with the results herein and include your comments in the main manuscript.
  • The introduction section should be enhanced with works (theoretical and experimental also in acidic environment) appeared the last 15 years concerning materials for EOR. (For example, see the works of G. Andreadis group, search in google scholar: ethanol + G. Andreadis).
  • The morphology/structure and the role of the active sites should be clearly pointed out in relation to the results obtained from physicochemical characterizations.
  • The XRD pattern of Ni/C indicates an amorphous structure since the Ni (111), (200), and (220) peaks are absent. Furthermore, the (200), (220), and (311) peaks presented in the XRD patterns of all the catalysts are significantly mild or even absent. Could these facts indicate the low electrochemical activity of the prepared catalysts? Could the synthesis method be applied insufficiently? Were the combined metals sufficiently alloyed? By the way, at which temperature did the metal precursors stirred with the NaBH4 in the synthesis step (high temperatures are generally required to achieve efficient alloying)?
  • The authors extracted a small average nanoparticle size for the three catalysts from the XRD and TEM measurements. However, the extensive agglomerates in the whole surface of all the catalysts that are apparent in the TEM images do not justify these results. Could the authors clearly explain this incompatibility?
  • What quantity of ethanol was added for the electrochemical experiments?
  • Was the right amount of metal precursors (Table 1) used to achieve the nominal molar ratio of the prepared electrocatalysts? If not, the discrepancy between the metal loadings extracted from EDX with the nominal one could be further explained.
  • Based on the XPS results, it is concluded that Ir exists only in metallic form, not presenting any oxidized species; this is not an expected outcome. Furthermore, why does Ni exist mainly in oxidized form? Was Ni alloyed sufficiently with the other two metals? If not, does this fact affect the electrochemical activity of the trimetallic catalysts? Furthermore, the authors should clearly explain how Ni could enhance the air stability of Pd and Ni.
  • Stability information of the material should be provided (included) in the main manuscript (one hour is not enough).
  • Some Table and Figure references in the manuscript are wrong. The manuscript must be double-checked for errors. Furthermore, reference [9] is presented twice in the manuscript (also as reference [15]).
  • The section for Results (& Discussions) should be better divided on the basis of the subject for the convenience of the Reviewers and Readers of Micromachines.
  • References should be according to the author guides and checked again (References [9] and [15] are same).
  • English grammar and syntax need improvement. The manuscript should be double-checked for carelessness mistakes.

Author Response

Please send the attachment.

Round 2

Reviewer 1 Report

The paper has been somewhat improved. However, some minor issues still need to be corrected before publication.

As I mentioned earlier:

  1. The legends to Fig. 4 and 6 have different fonts. Once it is big, once it is so tiny that barely visible.
  2. In Fig. 7, you have mV and V. Correct this.
  3. The authors should provide additional data on ethanol oxidation on the Ni surface for the examined electrode. It can be done as a supplementary file. However, it is crucial to prove their claims.

Author Response

Dear Colleague,

Thank you for making these comments about our manuscript. 

  • The legends to Fig. 4 and 6 have different fonts. Once it is big, once it is so tiny that barely visible.
  • Author Response: The legends are fixed now. 
  • In Fig. 7, you have mV and V. Correct this.
  • Author response: this is corrected now. 
  • The authors should provide additional data on ethanol oxidation on the Ni surface for the examined electrode. It can be done as a supplementary file. However, it is crucial to prove their claims.
  • Author response: Further articles are referred to and cited in the main text about EOR on Ni elecrodes at lines 292-294. 

Reviewer 2 Report

The authros took into consideration all my suggestions and appropriately revised their manuscript. I think now it could be accepted for publication to Micromacchines.

Author Response

Dear Colleague,

Thank you very much for taking the time to review our manuscript. 

Best wishes, 

The Authors